# Physicochemical Characteristics of Dual-Purpose Cow’s Milk During the Dry and Rainy Seasons in a Tropical Environment

**DOI:** 10.3390/vetsci12030269

**Published:** 2025-03-13

**Authors:** Maricela Ruiz-Ortega, Ethel Caterina García y González, Aurora Matilde Guevara-Arroyo, Alfonso J. Chay-Canul, Edgar Valencia-Franco, Marcos Pérez-Sato, José Vicente Velázquez-Morales, José del Carmen Rodríguez-Castillo, José Manuel Robles-Robles, Jorge Alberto Vázquez-Diosdado, José Luis Ponce-Covarrubias

**Affiliations:** 1Escuela Superior de Medicina Veterinaria y Zootecnia No. 3, Universidad Autónoma de Guerrero (UAGro), Técpan de Galeana 40900, Guerrero, Mexico; maricela_ruiz@uaeh.edu.mx (M.R.-O.); 09972@uagro.mx (A.M.G.-A.); 2Instituto de Ciencias Agropecuarias, Universidad Autónoma del Estado de Hidalgo, Tulancingo de Bravo 43600, Hidalgo, Mexico; 3División Académica de Ciencias Agropecuarias, Universidad Juárez Autónoma de Tabasco, Villahermosa 86280, Tabasco, Mexico; alfonso.chay@ujat.mx; 4Facultad de Ciencias Agrícolas y Pecuarias, Benemérita Universidad Autónoma de Puebla, Tlatlauquitepec 73900, Puebla, Mexico; edgar.valencia@correo.buap.mx (E.V.-F.); marcos.perez@correo.buap.mx (M.P.-S.); 5Instituto Mexicano del Seguro Social, Órgano de Operación Administrativa Desconcentrado, Hospital General de Zona No. 57, Cuautitlán 54769, Estado de México, Mexico; civentico@hotmail.com; 6Facultad de Medicina Veterinaria y Zootecnia, Benemérita Universidad Autónoma de Puebla, Tecamachalco 72570, Puebla, Mexico; jose.rodriguez@correo.buap.mx (J.d.C.R.-C.); manuel.roblesr@correo.buap.mx (J.M.R.-R.); 7School of Veterinary Medicine and Science, Sutton Bonington Campus, University of Nottingham, Leicestershire LE12 5RD, UK; Jorge.VazquezDiosdado@nottingham.ac.uk

**Keywords:** tropical dairy systems, seasonal variation, dual-purpose cattle, hot environment

## Abstract

In Mexico, the production of milk from dual-purpose cows in tropical conditions is important for the food security of rural families. The physicochemical composition of milk from dual-purpose cows was assessed during the rainy and dry seasons in the tropics. Milk sampled during the dry season had a higher percentage of fat and higher density. Milk fat and temperature were also higher during the dry season, and density, freezing point and conductivity were higher during the rainy season. Density was positively correlated with protein, freezing point and lactose. Fat was negatively correlated with density, freezing point, acidity and conductivity. These characteristics of milk were identified under the production system in a tropical environment in the state of Guerrero, providing the information necessary to propose management strategies in feeding, pasture management and animal welfare under these study conditions. Finally, this knowledge allows us to identify new marketing channels for the product, benefiting producers and their families in rural areas.

## 1. Introduction

At the global level, Mexico has a cattle inventory of 36.6 million heads and is ranked fifth (2,214,928 tons of meat), with 58.6% of the production for meat and dual purposes (meat and milk), and 4.6% for milk [1]. Within this context, the state of Guerrero has 1 million, 300 thousand heads of cattle, across 42,000 production units (PUs); of these, 16% are located in the Costa Grande region and 84% in the rest of the state [2]. The municipality of Atoyac de Alvarez, located in the Costa Grande region, is one of the most important in the production of milk from dual-purpose cows (24,024 heads) [3]. In fact, this municipality is ranked second in milk production (6,550 33 L per year) and that of its derivatives (cheese, cream and cottage cheese, among others), preceded by the municipality of Benito Juarez [3]. These products are marketed locally, in other municipalities in the region and in other states of the country.

The predominant bovine production system in the state of Guerrero is known as dual-purpose (meat and milk), based on grazing native and introduced plants, with little or no use of supplements, with a milking process with the calf at foot and with calves for sale at weaning [4,5]. The predominant genetic types are crosses of *Bos taurus* × *Bos indicus* [6], the labor is family-based and its profitability is low [7], causing many operational problems in the farms and families of producers.

In the dairy industry, raw cow’s milk is valued for its physical and chemical characteristics (water, fat, protein, lactose and minerals), characteristics that are modified by the effect of factors related to breed, age, interval between milkings, udder quarters, feeding, diseases, seasons of the year and ambient temperature [8,9]. For example, milk production is seasonal, increasing in the spring to summer months and declining in the fall and winter [10]. Physicochemical changes in milk occur according to the conditions in which dairy production takes place, directly impacting the environment and the availability and quality of pastures.

The physicochemical composition of cow’s milk is 4.0% fat, 3.6% protein, 5.0% lactose and less than 0.2% mineral salts, suspended in 87.7% of water [11]. Indeed, Pal et al. [12] report variations in fat and total solids contents between seasons, with the highest values obtained in winter, followed by the rainy seasons and summer. Additionally, Vega et al. [13] compared milk characteristics between different seasons of the year (rainy and dry), indicating differences in the variables pH, total solids and non-fat solids (NFSs), demonstrating that the milk samples analyzed changed from one season to another, which was possibly associated with changes in acidity in the forage during the rainy season. The information reported in the literature shows some findings on the change in the physicochemical characteristics of milk associated with a balanced diet during sampling. This study is important because it concerns dual-purpose cows in tropical conditions, emphasizing that cows consume forage in pastures, hence the marked change between seasons of the year.

For the above reasons, the aim of this study was to analyze the physicochemical composition of milk in dual-purpose cows during the rainy and dry seasons in a tropical environment in the state of Guerrero.

## 2. Materials and Methods

### 2.1. Study Area

The present study was carried out in the community of “El Ticui” municipality in Atoyac de Alvarez, Guerrero, Mexico. The site is located in the tropics at the geographical coordinates of 17°12′59″ north latitude and 100°26′40″ west longitude with respect to the Greenwich meridian. The average annual maximum temperature is 33 °C from April to September and the minimum is 17 °C from November to January (Figure 1). The rain occurs during the months of July to September (average annual rainfall is 2956 mm). All animal management procedures were conducted within the guidelines of nationally approved techniques for animal use and care [14]. The experimental protocol was approved on 28 March 2022 by the Committee for the Uses and Care of Experimental Animals of the Autonomous University of Guerrero (protocol #1028).

### 2.2. Sampling

To obtain the data, existing records of dual-purpose production units (PUs) in the municipality of Atoyac de Alvarez were consulted; both the producer and the workers were interviewed during visits made to the PUs. The questions were categorized by the number of animals, the number of milking cows and general management by season of the year (rainy vs. dry). In 10 herds of dual-purpose milking cows, a total of 237 milk samples were taken at different dates in the dry season (January and February; 128 samples) and rainy season (July and August; 109 samples). Each sample was taken for 3 consecutive days for each producer, ending with one and continuing with the other. Each sample was taken directly from the cow’s nipple (100 mL of milk) and placed directly in airtight sample jars, which were identified with the name of the producer and the cow number, and marked using white tape and an indelible black marker.

### 2.3. Laboratory Analysis

The samples were analyzed in the multidisciplinary laboratory of ESMVZ-3, UAGro with an Ekomilk Bond Total Ultrasonic milk analyzer (brand: EKOMILK^®^ Bond; Eon Trsding, Stara Zagora, Bulgaria). The milk analyzer has the capacity to measure 65 to 70 measurements per hour and provides results for each sample in <40 s. The variables evaluated by this equipment were fat (%), non-fat solids (NFSs; %), density (g/mL), freezing point (°C), acidity (%), temperature (°C), lactose (%), conductivity (mS/cm), pH and water (%).

### 2.4. Statistical Analysis

Descriptive data were calculated as the mean, standard deviation (SD) and coefficient of variation (CV) of the properties fat (%), NFS (%), density (g/mL), freezing point (°C), acidity (%), temperature (°C), lactose (%), conductivity (mS/cm), pH and water (%) using the PROC MEANS procedure. The data were analyzed through ANOVA, using the Kolmogorov–Smirnov test, confirming that the data fit a normal distribution. Additionally, one-way analyses of variance for each of the different characteristics were performed. The sampling season (rainy and dry) was considered as a fixed effect and the number of samples was considered as a covariate. The Tukey test was used with a significance level of *p* ≤ 0.05 to determine the differences in each parameter with respect to the season. Finally, a Pearson correlation was performed, from which a table was created with the coefficient value for the correlation between the properties. All statistical analyses were performed using the SAS 9.4 [15] program.

## 3. Results

In this study, a higher percentage of fat (5001 vs. 2975%) was found in the rainy season compared to the dry season (*p* < 0.05). Likewise, a higher density (1033.692 vs. 1035.45 g/mL) of milk was observed during the dry season compared to the rainy season (*p* < 0.05). The rest of the physicochemical characteristics of milk from dual-purpose cows in the tropics were similar between seasons (rainy vs. dry; *p* > 0.05) (Table 1).

Regarding the comparison of means by season of the year (rainy vs. dry), a significant difference was found in the variables of fat, density, freezing point, temperature and conductivity (*p* < 0.001). The milk temperature variable had a higher value in the dry season than in the rainy season; however, the density, freezing point and conductivity variables had higher values in the rainy season compared to the dry season (*p* < 0.001). In the variables of NFSs, protein, acidity, lactose, pH and water percentage, the results indicate that there was no significant difference between the two study seasons (*p* > 0.05) (Table 2).

Table 3 shows the results of the Pearson correlation coefficient (PCC) estimation for the variables with a statistically significant correlation. No differences were found in the PCC matrix due to the effect of the study season (rainy vs. dry), so the results of the PCC matrix are reported in a general way.

The variables of NFSs, protein and water maintained a correlation with all variables. The density maintained a positive correlation with the variables of protein, freezing point and lactose (*p* < 0.05). For its part, the fat variable presented a negative correlation with the variables of density, freezing point, acidity and conductivity (*p* < 0.05). On the other hand, the protein, freezing point, acidity and lactose variables presented 14% PCC. Finally, temperature was not correlated in any of the cases (*p* > 0.05) (Table 3).

Figure 2 shows the ambient temperature (max 30 °C), the relative humidity (max 70%) and the temperature and humidity index (THI; max 80%); during this study, higher values were observed during the rainy season compared to the dry season (*p* < 0.001). This indicates that dual-purpose cows in the tropics during the rainy season (summer) suffer heat stress, although this might be compensated for because this is the time of year when there is greater availability and quality of pastures.

## 4. Discussion

In this study, we found a higher fat percentage and density in milk samples taken during the rainy season compared to samples taken during the dry season. When comparing the averages by season of the year (rainy vs. dry), the variables of fat and milk temperature had higher values during the dry season than during the rainy season, while the variables of density, freezing point and conductivity were higher in the rainy season. The variables with the highest correlation were protein, freezing point, acidity and lactose. The variables of NFSs, protein and water were correlated with all milk characteristics. The density maintained a positive correlation with the variables of protein, freezing point and lactose. Finally, fat was negatively correlated with the variables of density, freezing point, acidity and conductivity.

The Liconsa Raw Milk Quality Control Standards Manual [16] defines milk as the normal secretion of the mammary glands of healthy cows, being a heterogeneous, white liquid, with a sweet taste and ionic reaction (pH) close to neutrality. This liquid must not contain substances foreign to its natural composition, such as bactericides, bacteriostatics, chemical or biological preservatives, antibiotics or toxic substances. The NMX-F-700 COFOCALEC-2004 specifies that whole cow’s milk should contain between 3.2 and 4.0% protein, 4.7 to 5.0% lactose and 3.4 to 6.0% fat [17]. Therefore, the milk from the PUs of the dual-purpose system of this research during the rainy season and the dry season is within the percentage of the standard, at 3.6%. In the case of NFSs, the results of the present study are above those indicated by the standard, with 9.7% in both seasons; the standard indicates 8.5% in a study carried out in the state of Chiapas, Mexico [18], and 8.7% SNG is reported in PUs, which is like the results of the present study. In other tropical countries with similar characteristics to those where the present study was carried out, values of 12.37% of total solids in Brazil and 13.95% in New Zealand are reported, as are values from continental countries such as Canada (12.97%). These higher values might appear because these countries have directed milk production towards the goal of obtaining higher total solids for significant yields in industrial processes [19,20]. In the present study, this characteristic of milk is outstanding since milk is used to make cheeses and this variable confers greater yield potential to the product, which is worth the increase in this characteristic. In the case of the fat variable, the results of the present study report a lower statistical value in the dry season, at 2.9%. In research carried out in the tropics in Ecuador in Pedernales and El Carmen, researchers obtained values of 2.6% and 2.7%, respectively [19].

This variation can be observed between cows of the same breed due to variation in diet and season of the year, since in this tropical region the forage consumed by cows is of better quality during the rainy season. Some of the factors that can potentially modify the percentage of fat in milk are the concentration of fiber in the diet and the ratio of forage or concentrate [21,22]. Therefore, the higher the fiber concentration, the higher the fat content in the milk due to the proportion of volatile fatty acids produced in the rumen [23,24]. This effect can be observed when comparing the two study seasons (rainy vs. dry), since in the rainy season the forage is of better quality, as has been reported in places in the same situation [13,25]. In the present study, no proximate chemical analyses were performed on the pastures regardless of the study season; it is important that in future research these chemical analyses be performed on the pastures to understand their nutritional characteristics. For example, Morales et al. [21] found that the fat content of milk presents significant differences due to the effect of the type of pastures in the study seasons (rainy vs. dry), emphasizing that the study was conducted in the Colombian tropics on Holstein cows under grazing conditions.

Tarqui-Mamani et al. [26] published the results for milk samples from the Majes sub-basin (characterized by a temperate desert climate) in Peru, with percentages of 3.29% for protein, 3.32% for fat and 8.91% for NFSs in 2007, and 3.26%, 3.24% and 9.01% in 2008. These results are similar to those found in the present study with dual-purpose cows from the tropics of Guerrero, Mexico.

The variable density of milk presented significant differences in the studied seasons (rainy vs. dry); in fact, the highest value was present in the dry season, at 35.4%. These values are not within the standard range (min 1029 and max 1033) established, and in fact they are higher than the established maximums. There is a study that explains that this is due to the lack of protein and energy sources in the diet [27]. The density of milk is important since it is an indicator of milk adulteration, mainly with the addition of water. Producers add a greater amount of water to increase the volume of their production and to deliver a greater amount of the product to the collection company, thus increasing the volume of milk and receiving a higher payment for the adulterated milk [19,28]. The pH varies during lactation. Indeed, the pH of colostrum is lower than that of milk, with a pH value of 6.0 indicating high protein content and pH values greater than 7.4 indicating the physiological state of lactation [29]. The results of the present study do not exceed these values. In this work, the characteristics of raw milk from dual-purpose cows were evaluated during the dry and rainy seasons. In both seasons, the pH of the milk was similar (6.8). In this state, normal pH value in milk shows that local producers have good management of this product, which does not contain contamination from impurities, infections or subclinical mastitis.

Regarding the physicochemical characteristics of milk from dual-purpose cows in the tropics of Veracruz, Mexico, Juárez-Barrientos et al. [30] identified the statistical relationship between milk variables, and mentioned that the lactose content was correlated with the SNG content (PCC = 0.82), added water (PCC = −0.81), density (PCC = 0.92) and freezing point (PCC = −0.90), while the SNG content was correlated with the added water content (PCC = −0.93), density (PCC = 0.94) and freezing point (PCC = −0.95). The results of the present study agree regarding the correlation of lactose with water (PCC = −0.41), the correlation of SNG with water (PCC = −0.35) and the correlation of density (PCC = 0.66) with freezing point (PCC = −0.61). The total solids content of milk is affected by the type of forage that the cattle receive [21,31]. Various investigations indicate that PCC values establish that the greater the presence of these variables, the greater the microbial growth; the physicochemical and compositional characteristics of milk offer the appropriate substrate for the growth of diverse microorganisms in which pathogens may be included, so it is important to know the PCCs [30,31,32].

Freezing point is an important indicator of milk quality that correlates with added water. For example, the freezing point in raw milk can be increased by factors such as high milk production, herd feeding and time of year [30,33]. The results obtained in the present study show that the seasons of the year had effects on the characteristics and composition of the milk, which can influence the price paid by the Mexican dairy industry to producers and highlight the importance of these factors in the quality of the milk.

In the case of conductivity, it varies according to the milking time, reaching values as high as 5.38 mS/cm [34]. In the present study, the conductivity values were lower than those indicated by this author, and we also found significant differences by season, with the higher value in the dry season (4605 mS/cm). Measuring the electrical conductivity of milk is also useful as a diagnostic method for intramammary infections, providing a diagnostic technique that is low-cost, is easy to determine and provides immediate results [35]. This variable shows that due to the genotype of cows used in the region and the low milk production compared to specialized dairy breeds such as Holstein, these cows also present fewer mammary gland diseases.

The physicochemical characteristics vary according to intrinsic factors (the breed, the individual and the lactation curve, among others) and extrinsic factors (feeding, milking and climatic conditions) affecting the cows [36]. Adverse effects have been observed in hot and humid climates in heat-stressed cows. Indeed, in arid regions of Mexico, an adverse effect on the physicochemical characteristics and an increase in the somatic cell count of milk during summer have been previously observed (see review [37]). On the other hand, Juárez-Barrientos et al. [30] found effects on the physicochemical quality of milk from dual-purpose cows in the tropics that are probably due to the type of feed; low-quality milk requires corrective actions focused on meeting the nutrient demands of the dairy herd, taking into account that in tropical conditions there is variability in the availability of forage and that the nutritional aspect is a key component for improving production in dual-purpose systems. It is necessary to consider forage factors by performing chemical–proximal analysis by season of the year (rainy vs. dry), since these directly influence the physicochemical characteristics of milk. It is also necessary to take into account the climate during the summer since it affects the thermoregulation of the cows and probably the characteristics of the somatic and microbiological cell count in the milk. To the authors’ knowledge, there are no reports in the literature on evaluations carried out under these study conditions in the Costa Grande region of the state of Guerrero, but there are reports on the thermoregulatory response of sheep [38,39]. Indeed, an increase in the THI during the summer (rainy season) affects the health of cows, which is reflected in a decrease in milk production.

Finally, returning to the climatic and feeding factors, as well as highlighting the adverse effect on dairy production caused in hot and humid tropical regions such as those in the state of Guerrero, more research is needed where these factors that influence the heat stress of cattle and the nutritional requirements of cows in milk production are considered. Indeed, it is important in tropical dual-purpose livestock production systems to evaluate the particular needs of each producer, which influence the production system and particularly the physicochemical and microbiological characteristics of the milk that directly impact the characteristics of the cheeses produced in the region. These are actions that would add value to dairy products and by-products and benefit local cattle and cheese producers.

## 5. Conclusions

It is concluded that the physicochemical characteristics of raw milk from dual-purpose cows per herd present better characteristics of fat, density, freezing point and conductivity during the rainy season. This might be because during the rainy season there is greater availability and quality of forage, which directly influences the productive state of the cows, reflected in the physicochemical quality of the milk. It is suggested that further research needs to be carried out on the characteristics of milk, considering the diet by season of the year and evaluating the chemical characteristics of the forage consumed by grazing cows and what is provided to them in the corral, especially during the dry season. This will improve dairy production and help producers make decisions about their herds, improving feeding aspects and taking into account heat stress and animal welfare. Likewise, decisions can be made regarding the nutritional management of lactating cows on pasture, seeking improved pasture strategies with a higher percentage of protein, which influences the quality of the milk regardless of the time of year.

## Figures and Tables

**Figure 1 vetsci-12-00269-f001:**
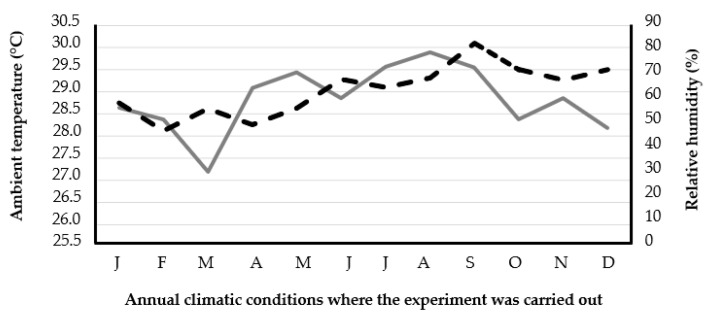
Annual behavior of ambient temperature (solid line) and relative humidity (dashed line) in the study area.

**Figure 2 vetsci-12-00269-f002:**
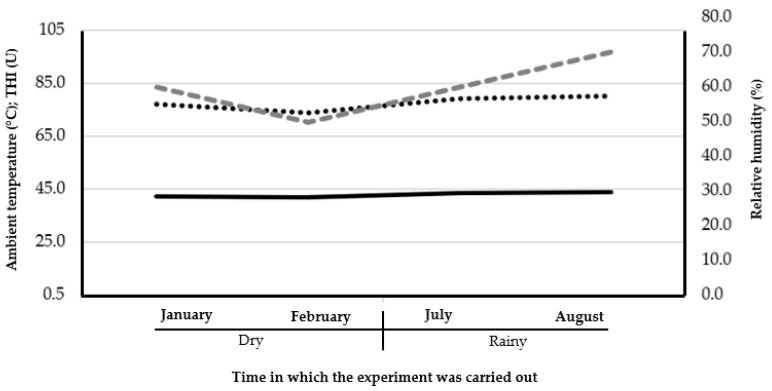
On the left “Y” axis, the ambient temperature (solid line) and THI (black dotted line) are observed. On the right “Y” axis, the relative humidity (gray dotted line) is observed. On the “X” axis, the evaluation months are observed according to the dry vs. rainy seasons.

**Table 1 vetsci-12-00269-t001:** Physicochemical characteristics of milk from dual-purpose cows in the tropics by evaluation season (rainy vs. dry).

Variable	Rainy Season				Dry Season			
	N	Average	SD	CV	N	Average	SD	CV
Fat (%)	109	5001 *	2446	48,904	128	2975	1949	65,503
Non-fat solids (%)	109	9748	0.656	6729	128	9777	0.584	5969
Density (g/mL)	109	1033.692	3354	9956	128	1035.45 *	2761	7789
Protein (%)	109	3682	0.254	6889	128	3655	0.219	5985
Freezing point (°C)	109	−0.581	4166	7171	128	−0.596	4088	6855
Acidity (%)	109	12,412	4279	34,472	128	11,756	3.91	33,257
Temperature (°C)	109	30,583	1169	3824	128	30,248	1.19	3933
Lactose (%)	109	5331	0.341	6404	128	5368	0.322	5997
Conductivity (mS/cm)	109	4392	0.371	8443	128	4605	0.606	13.15
pH	109	6891	0.02	1123	128	6882	0.0352	0.856
Water (%)	109	0.445	1594	358,148	128	0.396	1579	399,236

* within a row means there is a significant difference (*p* < 0.05).

**Table 2 vetsci-12-00269-t002:** Physicochemical characteristics of milk from dual-purpose cows by season of the year (rainy vs. dry; ANOVA).

Variable	Season		*p* Value
	Rainy	Dry	
Fat (%)	5001 ± 2446	2975 ± 1.949	<0.0001 **
Non-fat solids (%)	9748 ± 0.656	9777 ± 0.584	0.7175
Density (g/mL)	1033.692 ± 3354	1035.45 ± 2761	<0.0001 **
Protein (%)	3682 ± 0.254	3655 ± 0.219	0.3715
Freezing point (°C)	−0.581 ± 4166	−0.596 ± 4088	0.0048 *
Acidity (%)	12,412 ± 4279	11,756 ± 3.91	0.219
Temperature (°C)	30,583 ± 1169	30,248 ± 1.19	0.03 *
Lactose (%)	5331 ± 0.341	5368 ± 0.322	0.3968
Conductivity (mS/cm)	4392 ± 0.371	4605 ± 0.606	0.0016 *
pH	6891 ± 0.02	6882 ± 0.0352	0.1234
Water (%)	0.445 ± 1594	0.396 ± 1579	0.8106

** Highly significant difference (*p* < 0.0001). * Significant difference between means (*p* < 0.05).

**Table 3 vetsci-12-00269-t003:** Statistical relationships between the variables of physicochemical characteristics of milk from dual-purpose cows.

Statistical Relationship Between Variables		PCC ^1^	*p* Value
Fat	Density	−0.68495	<0.0001 **
Fat	Freezing point	−0.51373	<0.0001 **
Fat	Acidity	−0.15163	0.0195 *
Fat	Conductivity	−0.40072	<0.0001 **
Fat	Water	0.28233	<0.0001 **
Non-fat solids	Density	0.66338	<0.0001 **
Non-fat solids	Protein	0.79211	<0.0001 **
Non-fat solids	Freezing point	−0.71412	<0.0001 **
Non-fat solids	Acidity	−0.19766	0.0022 *
Non-fat solids	Lactose	0.80276	<0.0001 **
Non-fat solids	Conductivity	−0.22458	0.0005 *
Non-fat solids	pH	0.15899	0.0143 *
Non-fat solids	Water	−0.40563	<0.0001 **
Density	Protein	0.60431	<0.0001 **
Density	Freezing point	0.91188	<0.0001 **
Density	Lactose	0.69496	<0.0001 **
Density	Water	−0.51529	<0.0001 **
Protein	Freezing point	0.64512	<0.0001 **
Protein	Acidity	−0.18841	0.0036 *
Protein	Lactose	0.72356	<0.0001 **
Protein	Conductivity	−0.21884	0.0007 *
Protein	pH	0.17104	0.0083 *
Protein	Water	−0.3678	<0.0001
Freezing point	Acidity	−0.14136	0.0296 *
Freezing point	Lactose	0.76621	<0.0001 **
Freezing point	pH	0.12894	0.0474 *
Freezing point	Water	0.54045	<0.0001 **
Acidity	Lactose	−0.20617	0.0014 *
Acidity	Conductivity	0.14998	0.0209 *
Acidity	pH	−0.81087	<0.0001 **
Acidity	Water	0.17018	0.0087 *
Lactose	Conductivity	−0.21352	0.0009 *
Lactose	pH	0.17824	0.0059 *
Lactose	Water	−0.44137	<0.0001 **
Conductivity	pH	−0.13771	0.0341 *
Conductivity	Water	0.22068	0.0006 *
pH	Water	−0.17808	0.006 *

** Highly significant difference (*p* < 0.001). * Significant difference between means (*p* < 0.05). ^1^ PCC represents the Pearson correlation coefficient, and the variables fat, NFSs, protein, acidity, lactose and water are measured in percentages (%). The freezing point and temperature are measured in degrees Celsius (°C). Finally, density and conductivity are in g/mL and mS/cm, respectively.

## Data Availability

The original contributions presented in this study are included in the article. Further inquiries can be directed to the corresponding authors.

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
