# Peer review of "Physicochemical Characteristics of Dual-Purpose Cow’s Milk During the Dry and Rainy Seasons in a Tropical Environment"

_vetsci, 2025, doi:10.3390/vetsci12030269_

Round 1

Reviewer 1 Report

Comments and Suggestions for Authors

First of all, thank you for your submission. In your manuscript, through the study of the physical and chemical properties of milk from dual - purpose cows in the tropical region during the wet and dry seasons, you found that the physical and chemical properties of milk were better in the rainy season, providing an important basis for local dairy cow breeding management and industrial development. This is highly commendable. However, before your article can be accepted, there are still some issues that need to be addressed:
1.The title of the article mentions "tropical region", but most of the content in the article focuses on one area. Is it possible to compare with the research results of other tropical regions or regions with similar climates but different feed resources, and analyze the similarities, differences, and the underlying reasons?
2.The current sampling method may not fully represent the situation of the entire wet and dry seasons. Is it possible to reduce the error caused by intra - seasonal fluctuations by increasing the number of sampling time points?
3.The study should elaborate on the specific differences in various nutritional components in the feed of dairy cows in the rainy and dry seasons, and how these differences precisely affect the synthesis and metabolism of components such as fat and protein.
4.The study emphasizes the importance of feed on milk quality. A chemical analysis of the feed should be carried out to provide more direct evidence for explaining the changes in the physical and chemical properties of milk.
5.In the conclusion section, in addition to summarizing the research results, it should respond to the introduction part, expound on the practical guiding significance of these results for the local dairy cow breeding industry, and enhance the application value of the research.
6.For relationships like those in Table 3, is it possible to use graphs to illustrate them, which can facilitate readers to make more obvious comparisons?

Author Response

Referee 1

Review Report Form

Comments and Suggestions for Authors

First of all, thank you for your submission. In your manuscript, through the study of the physical and chemical properties of milk from dual - purpose cows in the tropical region during the wet and dry seasons, you found that the physical and chemical properties of milk were better in the rainy season, providing an important basis for local dairy cow breeding management and industrial development. This is highly commendable. However, before your article can be accepted, there are still some issues that need to be addressed:

  1. The title of the article mentions "tropical region", but most of the content in the article focuses on one area. Is it possible to compare with the research results of other tropical regions or regions with similar climates but different feed resources, and analyze the similarities, differences, and the underlying reasons?
  2. In the manuscript it is marked in red letters.

The present study is compared with works from other tropical regions of Mexico and other countries that share characteristics of the tropics of Guerrero, where the study was carried out. This is interesting since there is little information in tropical latitudes, especially in Mexico. In the tropics there is theoretically an abundance of water and pastures beyond the period where the rains are concentrated (June to September), however, in recent years due to the phenomenon of global warming the dry season in the region has become critical as the quantity and quality of forage decreases, which is mostly native forage similar to that which practically leaves the pastures without organic matter (pure soil and stones). This is why it is important to highlight this very interesting production system, which is different from other dual-purpose cattle systems where cattle are grazed on improved induced grass pastures and irrigated using technological systems, and the pastures remain green all year round. Additionally, the animals receive supplementary food, which is why the quality of their milk is different. On the other hand, in dairy farms the stable production system is different because they do not have problems in anything that has to do with feeding, since the cows receive balanced diets according to the physiological stage of these animals. Comparing both dairy production systems, everything is different due to the investment capital. In dual-purpose dairy farming, the investment is zero, since the producers are low-income, while in stabled dairy farming, the investment is greater, an investment that is quickly recovered by the marketing of the product. Throughout the manuscript, red letters mark where work from other tropical regions is compared with that of the present study.

2.The current sampling method may not fully represent the situation of the entire wet and dry seasons. Is it possible to reduce the error caused by intra - seasonal fluctuations by increasing the number of sampling time points?

A.- To ensure that the samples in this research were representative of the population, sampling was random, with each animal having the same probability of being included in the study. It should be noted that sampling bias may also occur if the sample is too small to represent the target population, this is because the area does not have a large number of cattle under experimental conditions, so this observation will be considered in future research. Thank you very much for your contributions to improve this manuscript.

3.The study should elaborate on the specific differences in various nutritional components in the feed of dairy cows in the rainy and dry seasons, and how these differences precisely affect the synthesis and metabolism of components such as fat and protein.

A.- In fact, it is important to delve into the nutritional components by time of year (rainy vs. dry) and how it affects the quality of milk (fat and protein). However, in the present study we believe that it does not apply since we explain later the reasons why we did not carry out the chemical analysis of the food, which, although it is crucial in this type of work, at least in this one it was not possible for us to do so. However, within the scope of this work and our possibilities, we explain the importance of the analysis and how feed influences the quality of milk. Although, as we explained, in our conditions the grass is native and the drought is long, there is not a high possibility of having pastures with grass that can be irrigated to support dual-purpose cows.

4.The study emphasizes the importance of feed on milk quality. A chemical analysis of the feed should be carried out to provide more direct evidence for explaining the changes in the physical and chemical properties of milk.

  1. For this study we did not have the possibility of financing to carry out the proximal chemical analysis of the forage and unfortunately, we do not have laboratories trained for this type of analysis. However, we believe that given the characteristics of this first study, it could be published in this way and emphasize what we explained, but in a future study it is important to carry out this procedure to see the characteristics of the native and improved forage by time of year, as well as the possibility of offering food supplements that improve the quality of the milk and integrate the characteristics of the artisanal cheeses that are made in this region since this is the use that is mostly given to milk. We are very grateful for the interesting recommendation you have made, as well as for the depth you can give to this type of study, considering the importance of carrying out these analyses since these characteristics of the food impact the quality of the milk.

  2. In the conclusion section, in addition to summarizing the research results, it should respond to the introduction part, expound on the practical guiding significance of these results for the local dairy cow breeding industry, and enhance the application value of the research.
  3. Okay, the manuscript highlights in red letters the practical importance for the local dairy industry, and the implications for local producers. Likewise, the importance of managing dairy cows on pasture by introducing high-protein pastures which influences the improvement of milk quality regardless of the time of year in which it is milked.

  4. For relationships like those in Table 3, is it possible to use graphs to illustrate them, which can facilitate readers to make more obvious comparisons?

A.- This research work is based on large amounts of data that can be easily summarized and read through tables. When writing the research paper, it is important that the data presented to the reader be specific in each of the correlations found in the results of the analysis because they are different and important in future research. The data in a graph is confusing, due to its size, the specification of the correlation of variable 1 and variable 2. For the above reasons, we request that the table be kept since the content is important and relevant.

Reviewer 2 Report

Comments and Suggestions for Authors

The manuscript investigates the physicochemical characteristics of dual-purpose cow's milk in a tropical environment across seasons, offering valuable insights for dairy management. While the study is relevant, several revisions are necessary to improve clarity, accuracy, and scientific rigor. Below are detailed recommendations:

  1. Title Correction

Issue: The term "Qual-Purpose" is incorrect. Replace with "Dual-Purpose Cow’s Milk" (Line 1).

  1. The abstract incorrectly states higher fat in the dry season, conflicting with Results (Table 1: rainy season fat = 5.001%, dry = 2.975%).Correct the abstract to reflect higher fat in the rainy season (Lines 20–23).
  2. Critical Data Errors

Density Units: Reported values (33.6–35.4 g/cm³) are implausible. Milk density is typically ~1.03 g/cm³. Likely a decimal error (e.g., 1.0336 vs. 1.0354). Verify and correct all density values (Tables 1–2, text).

Freezing Point: Values (58–59°C) are erroneous; milk freezes near -0.5°C. Check instrument calibration and correct to negative values (e.g., -0.58°C vs. -0.59°C).

  1. Sampling Methodology

No justification for selecting 10 herds or 100 samples. Potential selection bias. Clarify herd selection criteria (random/convenience) and sample size calculation (Lines 95–98, Methods).

  1. Statistical Model Clarification

The ANOVA model lacks details (e.g., fixed/random effects, herd as a factor). Specify the statistical model, including covariates and herd effects (Lines 133–138).

  1. Diet is cited as a factor but not analyzed.Acknowledge this limitation and recommend future studies to include forage proximate analysis (Lines 225–227, Discussion).

Invalid URLs (e.g., Reference 3) and incomplete citations. Verify all references and update broken links (References section).

Keywords are too generic (e.g., "cow’s milk"). Replace with terms like "tropical dairy systems," "seasonal variation," or "dual-purpose cattle" (Line 73).

  1. Inconsistent use of commas/periods (e.g., 5.001* vs. 33,692). Standardize to periods for decimals (Tables 1–2, text).
  2. Axis labels and legends in Figures 1–2 are unclear.Add units (e.g., °C, %) and improve resolution (Lines 102–106, 194–199).
  3. Limitations (e.g., lack of herd management data) are unaddressed.Include a brief limitations paragraph (Lines 312–317).
Comments on the Quality of English Language

Conditional Acceptance pending thorough revision of data presentation (density, freezing point), abstract correction, and methodological clarifications. The study’s contribution to tropical dairy science is notable, but accuracy and transparency must be ensured.

Author Response

Referee 2

Review Report Form

Comments and Suggestions for Authors

The manuscript investigates the physicochemical characteristics of dual-purpose cow's milk in a tropical environment across seasons, offering valuable insights for dairy management. While the study is relevant, several revisions are necessary to improve clarity, accuracy, and scientific rigor. Below are detailed recommendations:

  1. Title Correction

Issue: The term "Qual-Purpose" is incorrect. Replace with "Dual-Purpose Cow’s Milk" (Line 1).

  1. We agree, it was changed in the text and the letters are marked in red.

  1. The abstract incorrectly states higher fat in the dry season, conflicting with Results (Table 1: rainy season fat = 5.001%, dry = 2.975%). Correct the abstract to reflect higher fat in the rainy season (Lines 20–23).
  1. The fat results in the summary were changed according to the season (dry vs rainy), the letter is marked in red. Thank you very much for the observation.
  1. Critical Data Errors

Density Units: Reported values (33.6–35.4 g/cm³) are implausible. Milk density is typically ~1.03 g/cm³. Likely a decimal error (e.g., 1.0336 vs. 1.0354). Verify and correct all density values (Tables 1–2, text).

A.- The composition characteristics of milk include physical and chemical properties. Among the physical ones, there is the density that can be defined as the weight of a liter of milk expressed in kilograms, and it has been established that the density of raw milk at 15oC ranges between 1028 and 1034 g/ml. (Reference: https://doi.org/10.1016/B978-0-12-810530-6.00002-X)

Freezing Point: Values (58–59°C) are erroneous; milk freezes near -0.5°C. Check instrument calibration and correct to negative values (e.g., -0.58°C vs. -0.59°C).

A.- The values ​​that were held were confirmed, and indeed we made a typo. We modified it in the tables and the manuscript in general. Excellent observation, thank you very much. It is marked in the article with red letters.

  1. Sampling Methodology

No justification for selecting 10 herds or 100 samples. Potential selection bias. Clarify herd selection criteria (random/convenience) and sample size calculation (Lines 95–98, Methods).

A.- From each herd of dual-purpose cows, 10 animals were randomly selected for sampling per season, complying with animal health and welfare standards. To reduce sampling error, the collection was increased with 109 samples in the rainy season and 128 samples in the dry season. Thank you very much for the observation, it is marked in red in the sampling of the materials and methods section.

  1. Statistical Model Clarification

The ANOVA model lacks details (e.g., fixed/random effects, herd as a factor). Specify the statistical model, including covariates and herd effects (Lines 133–138).

A.- The sampling season (rainy or dry) was considered as a fixed effect, and the number of milkings was considered as a covariate. For this study, the effect of the number of births or the age of the mother was not considered.

  1. Diet is cited as a factor but not analyzed. Acknowledge this limitation and recommend future studies to include forage proximate analysis (Lines 225–227, Discussion).

A.- It is marked in red in the manuscript. It was modified, emphasizing that at least for this work it was not possible to measure the chemical characteristics of the forage but that for other upcoming works it can be done.

Invalid URLs (e.g., Reference 3) and incomplete citations. Verify all references and update broken links (References section).

A.- Verified and corrected in the manuscript. Marked in red.

Keywords are too generic (e.g., "cow’s milk"). Replace with terms like "tropical dairy systems," "seasonal variation," or "dual-purpose cattle" (Line 73).

  1. Keywords were replaced as recommended.
  1. Inconsistent use of commas/periods (e.g., 5.001* vs. 33,692). Standardize to periods for decimals (Tables 1–2, text).

A.- It was standardized by points in the decimals according to the recommendation, it is marked in red, thank you very much.

  1. Axis labels and legends in Figures 1–2 are unclear. Add units (e.g., °C, %) and improve resolution (Lines 102–106, 194–199).

A.- Figures 1 and 2 have been modified for comprehension, taking into account labels and legends. They are marked in red in the manuscript.

Limitations (e.g., lack of herd management data) are unaddressed. Include a brief limitations paragraph (Lines 312–317).

A.- A paragraph was written addressing the limitations in the management of tropical dairy cattle herds. It is marked with red letters in the manuscript.

Comments on the Quality of English Language

Conditional Acceptance pending thorough revision of data presentation (density, freezing point), abstract correction, and methodological clarifications. The study’s contribution to tropical dairy science is notable, but accuracy and transparency must be ensured.

Dear Sirs, we thank you for taking the time to read our manuscript and advise us on your specialty, especially for the excellent comments made that served to improve the clarity and quality of the article.

Round 2

Reviewer 1 Report

Comments and Suggestions for Authors

The article has been greatly improved, it is recommended to accept it

Comments on the Quality of English Language

I don't care about language 

Reviewer 2 Report

Comments and Suggestions for Authors

The manuscript has been revised properly.